# Surfactant-Mediated Ultrasonic-Assisted Extraction and Purification of Antioxidants from *Chaenomeles speciosa* (Sweet) Nakai for Chemical- and Cell-Based Antioxidant Capacity Evaluation

**DOI:** 10.3390/molecules27227970

**Published:** 2022-11-17

**Authors:** Fuxia Hu, Feng Li, Zhenjia Zheng, Dongxiao Sun-Waterhouse, Zhaosheng Wang

**Affiliations:** 1Key Laboratory of Food Processing Technology and Quality Control in Shandong Province, College of Food Science and Engineering, Shandong Agricultural University, Taian 271018, China; 2School of Chemical Sciences, The University of Auckland, Auckland 1142, New Zealand

**Keywords:** *Chaenomeles speciosa*, phenolics, extraction, purification, antioxidant potential

## Abstract

In this study, a surfactant-mediated ultrasonic-assisted process was used for the first time to produce an antioxidant-enriched extract from *Chaenomeles speciosa* (Sweet) Nakai (*C. speciosa*, a popular fruit grown widely in the temperate regions of China). Ultrasonic treatment at 51 °C and 200 W for 30 min with sodium dodecyl sulfate as the surfactant led to a phenolic yield of 32.42 mg/g from dried *C. speciosa* powder, based on single-factor experiments, the Plackett–Burman design and the Box–Behnken design. The phenolic content increased from 6.5% (the crude extract) to 57% (the purified extract) after the purification, using LSA-900C macroporous resin. Both the crude and purified extracts exhibited a significant total reducing power and DPPH/ABTS scavenging abilities, with the purified extract being more potent. The purified extract exerted significant antioxidant actions in the tert-butyl hydroperoxide-stimulated HepG2 cells, e.g., increasing the activities of superoxide dismutase and catalase, while decreasing the reactive oxygen species and malondialdehyde levels, through the regulation of the genes and proteins of the Nrf2/Keap1 signaling pathway. Therefore, the extract from *C. speciosa* is a desirable antioxidant agent for the oxidative damage of the body to meet the rising demand for natural therapeutics.

## 1. Introduction

Oxidative stress can arise from the excessive production of reactive oxygen species (ROS), including various free radicals and highly reactive ions, relative to the antioxidant defense, causing the accumulation of ROS in cells, oxidative damage to tissues, and the development and progression of associated diseases, such as liver, cardiovascular and neurodegenerative diseases [1]. The nuclear factor erythroid-2 related factor 2 (Nrf2)-Kelch sample related protein-1 (Keap1) signaling pathway, as the fundamental regulator of the cytoprotective responses to oxidative and electrophilic stress, can operate as a switch to resist cell death and play a crucial role in maintaining the cellular redox homeostasis and the body’s defense against various oxidative damages [2]. Nrf2, as a master regulator of cellular oxidative levels against environmental stresses can bind, along with small Maf proteins, to the antioxidant response element (ARE) of the target genes, whilst Keap1 can attach to Nrf2 then promote its degradation by the ubiquitin proteasome pathway [3]. Therefore, the Nrf2-Keap1 signaling pathway is a pharmacological target for the development of therapies counteracting oxidative damage, and the direct or indirect activation of this signaling pathway is essential to preventing and treating oxidative stress-related diseases.

Dietary antioxidants have been highly sought from natural resources, as at proper doses and in appropriate combinations, they can inhibit harmful oxidants, provide certain protection against cellular damage and organ dysfunction, and maintain the body performance [4]. To discover and recover the dietary antioxidants naturally occurring in the plants, a range of extraction methods have been developed, including those conventional processes (e.g., decoction extraction, solvent extraction and Soxhlet extraction) and emerging extraction techniques (ultrasonic-assisted extraction, microwave-assisted extraction, high pressure-assisted extraction, supercritical fluid extraction) [5,6,7,8]. Compared with conventional extraction methods, the emerging methods exhibit a number of advantages, including the extraction efficiency, absence of organic solvent (s), a low energy consumption, a low cost and environmental friendliness. The ultrasonic-assisted extraction is an efficient, green and inexpensive extraction method, which allows the mechanical energy, in the form of ultrasound waves, to evenly pass through a liquid medium containing solid particles, causing localized high temperatures (up to 4500 °C) and pressures (up to 50 MPa), thereby disrupting the cell membranes and facilitating the extraction of intracellular components [8,9]. Furthermore, the surfactant-mediated extraction has also been recognized as a versatile, green, low toxicity and inexpensive method for extracting bioactive substances from plant-based materials. Using surfactants with an appropriate hydrophilic-lipophilic balance value, leads to a high extraction yield and recovery of antioxidants, including phenolic compounds e.g., non-ionic surfactants might improve the antioxidant entrapment efficiency whilst the amphiphilic surfactants could enable the formation of micelles and assimilate both the polar and non-polar residues of phenolics [10,11]. The combined use of the surfactant-mediated extraction and ultrasonic-assisted extraction (termed “surfactant-mediated ultrasonic-assisted extraction”) may provide dual advantages, in terms of the increased extraction effectiveness (owing to the wetting and solubilizing effects of the surfactants, along with the mechanical, cavitation and thermal effects of the ultrasonic waves), as well as the enhanced energy efficiency, time saving, operational safety and environmental protection. Such advantages would be achieved through optimizing both processes to tailor the parameters affecting the extraction outcome, such as extraction time, extraction temperature, extraction pressure, the number of extractions, and ratio of the extraction medium to the raw material.

*Chaenomeles speciosa* (sweet) Nakai (*C. speciosa*, Rosaceae family) is rich in antioxidative substances (e.g., polyphenols), besides nutrients [12]. This medicinal plant exhibits various pharmacological activities, such as antioxidant, anti-inflammatory, immunoregulatory, antimicrobial, hepatoprotective, anti-cancer and analgesic effects [13]. Accordingly, it is of interest to develop effective methods for preparing products rich in antioxidants from *C. speciosa*. In this study, a crude antioxidant extract was produced via a surfactant-mediated ultrasonic-assisted extraction process before the purification using macroporous resins. Both the extraction process and the purification process were optimized. Then, the antioxidant capacities of the resulting unpurified (crude) and purified antioxidant extracts were evaluated by the total reducing power, the DPPH scavenging ability and the ABTS scavenging ability assays. The antioxidant actions of the purified extract were further investigated in a tert-butyl hydroperoxide (TBHP)-stimulated HepG2 cell model. The cell-based studies confirmed the significant antioxidant potential of the purified *C. speciosa* extract.

## 2. Results and Discussion

### 2.1. Effects of the Different Factors on the Extraction Yield of the Total Phenolics

As shown in Figure 1a, the use of each of the five selected surfactants increased the yield of the phenolic compounds from *C. speciosa*. Among the five surfactants, SDS led to the highest phenolic yield, probably due to the fact that SDS is an anionic surfactant, causing a greater solubilizing effect within the micelle, compared with the cationic and non-ionic surfactants [14]. As shown in Figure 1b, the highest phenolic yield occurred at a SDS concentration of 0.4 g/L, within the analyzed SDS concentration range of 0.2–1.2 g/L. It might be that a higher SDS concentration could accelerate the surface tension of the solvent and the interfacial tension between the solid and the liquid decrease, which made the solvent more easily infiltrate the powder of *C. speciosa*, thus increasing the polyphenol yield. When the concentration was above 0.4 g/L, due to the formation of micelle, its surface tension would not change with the increase of its concentration, and some impurities were dissolved, so the yield decreased [15]. Accordingly, the optimal SDS concentration was selected as 0.4 g/L.

In the ethanol concentration range of 30–80%, the phenolic yield reached the highest at an ethanol concentration of 60% (Figure 1c). This result was in close association with the polarity of the aqueous ethanol solutions, as well as the abilities of a highly concentrated ethanol to shrink/collapse the plant tissues, while denaturing and/or precipitating the proteins, causing it to block the tissue pores in *C. speciosa* (thereby decreasing the release of the phenolic compounds) [16,17]. The ethanol concentration of 60% was used for the subsequent experiments.

As shown in Figure 1d, the phenolic yield was significantly (*p* < 0.05) influenced by the material-to-liquid ratio, and it peaked at the ratio of 1:30 g/mL. These results indicate that a higher ratio of solvents could accelerate the infiltration of the solvent into the *C. speciosa* cells, thereby facilitating the extraction of the phenolic compounds [18]. However, the extraction yield changed slightly when the material-to-liquid ratio was less than 1:30 g/mL, because the continued increase of the solvents promoted the dissolution of the pigments and other substances, affecting the extraction effect of the phenolics [18]. In consideration of the solvent recovery and cost, the material-to-liquid ratio of 1:30 g/mL was used in the subsequent experiments.

The effects of the different ultrasonic conditions on the phenolic yield of *C. speciosa* polyphenols are shown in Figure 1e–g. The maximum phenolic yield was obtained at 50 °C for 30 min with an ultrasonic power of 200 W. As the ultrasonic temperature, power and time increased, the phenolic yield initially increased then decreased. These results might be due to the fact that a higher temperature, power or time led to (i) the oxidation and degradation of some phenolic compounds [18,19] and/or (ii) the reduction or partial inactivation of the surfactant activity, causing a decrease in the phenolic yield.

### 2.2. Optimization for the Extraction of the Phenolics

#### 2.2.1. Plackett–Burman Experiment Analysis

Based on the results of the single-factor experiments, a Plackett–Burman experimental design was performed to analyze the significance of each factor (Table 1 and Table 2). The *p* value of the model was 0.0051, indicating that the model was extremely significant and suitable for the prediction within the range of the variables employed. Meanwhile, the *p* values of the ethanol concentration, the material-to-liquid ratio and the ultrasonic temperature were all lower than 0.05, indicating that these three factors had significant influences on the extraction of the total phenolics. The ethanol concentration and the material-to-liquid ratio can directly affect the solubility of the phenolics in the solvent to promote the dissolution of the phenolics, so they had a significant influence on the extraction of the total phenolics [17,20]. The ultrasonic temperature, not only affected the oxidation and degradation of the phenolics, but also affected the activity of the surfactants. So, it had a significant influence on the ultrasonic time and power on the extraction of the total phenolics [21]. Therefore, the ethanol concentration, the material-to-liquid ratio and the ultrasonic temperature were used for the subsequent experiments.

#### 2.2.2. Response Model Establishment and the Variance Analysis

According to the Box–Behnken design principle, a three-factor (i.e., ethanol concentration, material-to-liquid ratio and ultrasonic temperature) response surface test was designed, and the results of the test results are shown in Table 3 and Table 4. Following the regression analysis using the Design Expert, the quadratic model for the phenolic yield and various factors was obtained:
Y (mg/g) = 3.24 − 0.073X_2_ + 0.046X_3_ + 0.045X_4_ − 0.013X_2_X_3_ − 0.019X_2_X_4_ − 6.750 × 10^−3^X_3_X_4_ − 0.18X_2_^2^ − 0.041X_3_^2^ − 0.18X_4_^2^
(1)

where: Y was the phenolic yield (mg/g); X_2_, X_3_ and X_4_ were the coded variables ethanol concentration (%), material-to-liquid ratio (g/mL), ultrasonic temperature (℃), respectively.

As shown in Table 4, the model was extremely significant (*p* < 0.001). The F value of the lack-of-fit item was 1.31 (*p* = 0.3861 > 0.05), suggesting that the model fits well and can explain the variation in the response. Meanwhile, the obtained regression coefficient and the adjusted coefficient of determination of the model (R^2^ = 0.9840, R^2^_Adj_ = 0.9634), provides a better fit and can explain 96.34% of the total changes in the response values.

#### 2.2.3. Response Surface Analysis

Figure 2 shows the interactive effects of the three test variables on the phenolic yield. The steeper the slope of the response surface, the more significant the change of response value, with a flat slope indicating an unobvious effect. [22]. In this study, the interaction terms (X_2_X_3_, X_3_X_4_, X_2_X_4_) were not significant (*p* > 0.05), the independent factors (X_2_, X_3_, X_4_) had a significant effect on the yield of the polyphenols (*p* < 0.05), which was consistent with the analysis of variance.

#### 2.2.4. Optimal Conditions and the Model Validation

Based on the above results, the optimal extraction conditions for the total phenolics from *C. speciosa* were obtained as follows: the SDS concentration, 0.40 g/L; ethanol concentration, 57.69%; the material-to-liquid ratio, 1:32.96 g/mL; the ultrasonic treatment, 51.24 °C, 200 W, 30 min. Under these conditions, the maximum theoretical phenolic yield was 32.66 mg/g. Considering the feasibility of the practical operation, the process parameters were further adjusted: the SDS concentration, 0.40 g/L; the ethanol concentration, 58%; the material-to-liquid ratio, 1:33 g/mL; the ultrasonic treatment, 51 °C, 200 W, 30 min. Under these conditions, the phenolic yield was 32.42 ± 0.27 mg/g (*n* = 3), indicating that the model was valid for predicting the phenolic yield.

### 2.3. Effect of the Different Factors on the Purification of the Phenolic Extract

#### 2.3.1. Static Adsorption and Desorption

Table 5 compares the effects of the different types of macroporous resins on the adsorption and desorption of the phenolic extract. Among the five test resins, LSA-900C and LSA-900E had higher adsorption capacities and adsorption rates than the other resins, (among which no significant differences were detected). However, LSA-900C had a significantly (*p* < 0.05) higher desorption rate and recovery rate than LSA-900E. Accordingly, the LSA-900C resin was more suitable for the purification of the total phenolics of *C. speciosa*.

It was reported that the pH of the solution affected the adsorption and the intermolecular force between the target component and the macroporous resin [23]. In this study, the adsorption ratio reached a maximum value of 94.15% at pH 3.0 (Figure 3a). As the pH further increased, the adsorption ratio continuously decreased, with the minimum value (54.41%) observed at pH 7.0. These changes might be attributed partially to the phenolic hydroxyl groups of the phenolic compounds present in the extract under acidic conditions, which facilitated a greater adsorption by the macroporous resin. As the pH increased, the solubility of the total phenolics increased, which weakened the interaction of the phenolics with the resin. Therefore, the optimal pH value was 3.

#### 2.3.2. Dynamic Adsorption and Desorption

As shown in Figure 3b, the concentration of the total phenolics in the eluate increased as the test sample concentration increased. If the sample concentration (4.0 mg/mL) is excessively high, the blockage of the resin may occur, which would negatively affect the adsorption of the polyphenols by the LSA-900C resin [24]. The leakage point and the saturation point appeared late when the sample concentration was 0.5 mg/mL, as the time for the elution along the upper column operation was too long. The leakage points appeared around 140, 90 and 60 mL for the sample concentrations of 1.0, 2.0 and 3.0 mg/mL, respectively. While the sample concentration was 2.0 mg/mL, the adsorption capacity and adsorption rate (347.47 mg and 77.7%, respectively) were higher than those when the sample concentrations were 1.0 and 2.0 mg/mL. Based on the results, the sample concentration of 2.0 mg/mL was selected for the subsequent experiments.

The effect of the elution flow rate on the adsorption performance of the LSA-900C resin is shown in Figure 3c. The higher the elution flow rate, the higher the content of the phenolics in the eluate. When the flow rate was 2.0 or 2.5 mL/min, the phenolics might be eluted out of the column without being fully adsorbed by the resin, resulting in the leakage points appearing at 40 and 50 mL and the adsorption rates being lower than 70%. Although the phenolics could be fully diffused and adsorbed by the resin when a lower elution flow rate was used, a longer adsorption time period would be required. Hence, the optimal elution flow rate was set at 1.0 mL/min and the sample volume was selected as 100 mL.

As shown in Figure 3d, the area of the desorption peak corresponding to 40% ethanol was significantly smaller with the peak being broader, compared with those corresponding to 50% and 60% ethanol. The desorption rates were 72.06% for 40% ethanol, 77.94% for 50% ethanol, and 75.46% for 60% ethanol. The bigger and narrower desorption peaks for the elution with 50% or 60% ethanol indicated the higher effectiveness of the desorption of the phenolics from the resins and a greater enrichment of the phenolics in the eluate. Moreover, the tailing phenomenon was not obvious in the case of 50% ethanol, indicating a relatively fast mass transfer in the particles and the complete desorption of the adsorbed molecules when 50% ethanol was used. Based on these results, a 50% ethanol solution was selected, which was consistent with the result of the static desorption experiments. A very low concentration of ethanol could not effectively disrupt the hydrogen bonds between the phenolics and the resins; The polarity of an aqueous ethanol would decrease with the increase of the ethanol concentration in water, causing a reduced desorption rate, due to the decreased solubility of the highly polar phenolic compounds (e.g., phenolic acids) in the ethanol-water eluent [25]. Our most recent study using HPLC-Q-TOF-MS/MS revealed that chlorogenic acid (71.964 mg/g dried extract), procyanidin B1 (43.773 mg/g dried extract), catechin (25.962 mg/g dried extract) and two procyanidin trimers (in trace amount) were present in the extract obtained from *C. speciosa* by 58% ethanol (*v*/*v*, in water), under the extraction conditions of the material-to-liquid ratio as 1/33 g/mL and the ultrasonic pretreatment, at 51 °C and 200 W for 30 min [26]. The very high ethanol concentration for the elution during the desorption process seemed not suitable for the extract of *C. speciosa* with the high presence of chlorogenic acid.

As shown in Figure 3e, when the elution flow rate was 2.0 or 2.5 mL/min, the peaks were relatively flat and broad with a significant tailing phenomenon. When the flow rate was too high, the eluent ran very fast, thereby reducing the dissolution of the phenolics in the eluent. At the elution flow rates of 0.5, 1.0 and 1.5 mL/min, the peak shapes of the elution curves were similar (relatively narrow and symmetric), indicating better elution effects. Considering the elution time and the amount of ethanol used (from the perspectives of both operational safety and cost effectiveness), the elution flow rate of 1.5 mL/min was more appropriate, and thus selected. The eluting volume was set as 120 mL, based on the elution curves.

Taken together, the optimum purification conditions for purifying the antioxidant-enriched extract were: the LSA-900C macroporous resin; the sample concentration, 2.0 mg/mL; pH, 3; the loading flow rate, 1.0 mL/min; the loading volume, 100 mL; the ethanol concentration, 50%; the elution flow rate, 1.5 mL/min; the eluting volume, 120 mL. Following the purification under such conditions, the content of the phenolics in the extract increased from 6.49% to 57.00%, indicating that the purification allowed the enrichment of the phenolics in the purified *C. speciosa* antioxidant-enriched extract.

### 2.4. Antioxidant Activities of the C. speciosa Antioxidant-Enriched Extract

#### 2.4.1. Antioxidant Activity Assay

The total reducing power, DPPH and ABTS methods, based on different mechanisms, are often used to evaluate the in vitro antioxidant capacities of a food product. In this study, these assays were conducted to provide a rapid and preliminary evaluation of the antioxidant potential of the purified *C. speciosa* extract.

As shown in Figure 4a, all test samples showed a dose-dependent increase in the total reducing power, with the total reducing power decreasing in the order of ascorbic acid > the purified antioxidant-enriched extract > the crude antioxidant-enriched extract. The increase in the reducing power caused by the purification of the extract was also reported in a previous study on the flavonoid-enriched extract from *Crocus sativus L.* (saffron) by Chen, Xiang, Liu, Li, and Yang [27]. For both DPPH and ABTS, the scavenging rates of all test samples increased with an elevated sample concentration, then tended to level off (Figure 4b,c). The DPPH and ABTS scavenging abilities of the purified extract were comparable to those of ascorbic acid. When the sample concentration was 40 µg/mL, the scavenging rates of the purified antioxidant-enriched extract towards DPPH and ABTS radicals reached 82.11% and 94.46%, respectively, which were significantly (*p* < 0.05) higher than those of the crude antioxidant-enriched extract. The EC_50_ values for the purified extract were calculated as 8.67 µg/mL (DPPH) and 9.97 µg/mL (ABTS). The catechin from *Syzygium cumini* seed kernel was found possessing a strong DPPH radical scavenging activity with an EC_50_ value of 5.35 µg/mL [28], which was comparable to the EC_50_ values of this study. Taken together, these results suggest that both the purified and crude antioxidant-enriched extracts had significant in vitro scavenging abilities, with the purified extract being more potent than the crude extract. Similar results were also found in the study on the total polyphenols from honeysuckle [29]. A previous study on different *Chaenomeles* species (although not *C. speciosa*) revealed that the extracts from fresh fruits, e.g., those from *C. sinensis* (containing phenolics, such as epicatechin and procyanidin B2) and *C. japonica* (containing phenolics, such as Catechin and procyanidin B1), exhibited strong ABTS and FRAP abilities [30].

#### 2.4.2. Effects of the Purified *C. speciosa* Antioxidant-Enriched Extract on the Viability of HepG2 Cells

To examine the potential cytotoxicity of the purified *C. speciosa* antioxidant-enriched extract, TBHP and quercetin (as a reference), the CCK-8 assay was performed to determine cell viability. As shown in Figure 5a, the relative viability of the cells treated with the purified extract at 10 µg/mL or lower, was higher than 90%, indicating that the purified extract at a concentration not higher than 10 µg/mL, did not have obvious toxic effects on the cells. Accordingly, the concentrations of 2.5, 5 and 10 µg/mL were used for the subsequent intervention experiments.

Figure 5b shows the cytotoxic effect of TBHP on the HepG2 cells. The TBHP treatment significantly (*p* < 0.05) decreased the cell viability in a dose-dependent fashion, with the IC_50_ value up to 499.1 µmol/L. Therefore, a TBHP concentration of 500 µmol/L was used to treat the cells to establish an oxidative damage model. As shown in Figure 5c, compared with the control, the treatment with TBHP at 500 µmol/L significantly (*p* < 0.05) decreased the relative viability of the cells, and the pre-treatment of the cells with the purified antioxidant-enriched extract counteracted significantly (*p* < 0.01) the TBHP-induced changes. Such a counteracting effect was in a dose-dependent fashion. These results indicated that the purified *C. speciosa* antioxidant-enriched extract can effectively protect the HepG2 cells against oxidative damage.

#### 2.4.3. Effects of the Purified *C. speciosa* Antioxidant-Enriched Extract on the Oxidative Stress Biomarkers

While the responses of cells to oxidation stress vary with initial stimulus/stimuli, the location of the occurrence and the cell’s health status, it was found that the activation of the oxidation pathways, the release of oxidation markers (e.g., ROS and MDA), and the increases in the levels of oxidation-related enzymes (e.g., SOD, CAT and GPx), commonly take place [31]. In addition to the ROS level, the content of MDA (a biomarker of oxidative stress) is widely used as an indicator for the lipid peroxidation. As shown in Figure 5d,e, the treatment with TBHP alone stimulated remarkably the production of ROS and MDA, whilst the presence of the purified *C. speciosa* antioxidant-enriched extract at 0.5, 5, or 10 µg/mL significantly suppressed such TBHP-induced increases of the ROS and MDA levels (such suppression was in a dose-dependent manner). Among the used doses of the purified extract, the dose of 10 µg/mL *C. speciosa* antioxidant-enriched extract led to the greatest antioxidant effect i.e., the largest decreases in ROS production (which was very similar to the effect of quercetin at 5 µg/mL; decreased by 58.97% for the purified extract and 60.14% for quercetin, respectively) (Figure 5d), and the largest decreases in the MDA production (which was very similar to the effect of quercetin at 5 µg/mL; decreased by 57.11% for the purified extract and 51.89% for quercetin, respectively) (Figure 5e).

SOD and CAT are the main antioxidant enzymes in mammalian cells. SOD scavenges superoxide free radicals through catalyzing the disproportionation of superoxide anion radicals into H_2_O_2_ and molecular oxygen, whilst CAT converts H_2_O_2_ into water, thereby protecting cells against oxidative damage [32]. As shown in Figure 5f,g, the TBHP stimulation reduced the activities of SOD and CAT in the cells by 2.25-fold and 1.69-fold, respectively (*p* < 0.01), compared with the control. The treatment of cells with the antioxidant-enriched extract significantly (*p* < 0.01) counteracted the TBHP-induced decrease in SOD and CAT activities, and such counteracting effects were in a dose-dependent manner. The activities of SOD and CAT in the cells treated with the purified antioxidant-enriched extract at 10 µg/mL and quercetin at 5 µg/mL, were 40.66 and 48.84 U/mg prot, and 36.55 and 47.05 U/mg prot, respectively. Similar antioxidant actions of polyphenols were reported previously. For example, the polyphenols from the *T. indica* seed alleviated oxidative damage through reducing the production of ROS and MDA, while increasing the enzyme activities of SOD, CAT and GPx in H_2_O_2_-induced HepG2 cells [33]. Therefore, the antioxidant actions of the purified *C. speciosa* antioxidant-enriched extract included the inhibition of the production of ROS and MDA, as well as the increases of the activities of SOD and CAT enzymes, by which HepG2 cells were protected against the oxidative damage caused by TBHP.

#### 2.4.4. Effects of the Purified *C. speciosa* Antioxidant-Enriched Extract on the Nrf2/Keap1 Pathways in TBHP-Induced HepG2 Cells

Nrf2/Keap1, as a key pathway for cells in response to cellular oxidative stress, induces/regulates the expressions of oxidative mediators (e.g., ROS, MDA) and oxidative-related enzymes and genes (e.g., SOD, CAT, GSH-Px, HO-1). In the resting/normal state, Nrf2 is combined mainly with its inhibitor, Keap1, and exists in an inactive state, thereby allowing a low transcription activity. When cells/cell-surface receptors are stimulated (e.g., upon ROS overproduction), Nrf2 and Keap1 are detached, then the disassociated and activated Nrf2 enter the nucleus to interact with the antioxidant response element (ARE) and regulate the expressions of the Nrf2 target genes and downstream target proteins, such as heme oxygenase-1 (HO-1) [34]. In this study, the influence of the purified *C. speciosa* antioxidant-enriched extract on the Nrf2/Keap1 pathway under the oxidative stress was examined.

As shown in Figure 5h, the expressions of Nrf2 and HO-1 proteins in the cells exposed to TBHP were significantly reduced by 40.59% and 46.58% respectively, compared with the control. The treatment with the purified extract, along with TBHP, led to significantly higher expressions of Nrf2 and HO-1, compared to the TBHP treatment alone, with the extract at 10 µg/mL, resulting in the greatest increases (27.95% and 35.58% for Nrf2 and HO-1, respectively). The stimulation of TBHP alone led to a 1.72-fold increase in the expression of the Keap1 protein, compared with the control, and the presence of the purified antioxidant-enriched extract effectively suppressed such a TBHP-induced increase, with the extract at 10 µg/mL being the most effective (suppressed by 40.89% for the Keap1 expression). Similar antioxidant actions of polyphenol-rich extracts were reported previously, for example, the extract from *Tussilago farfara L.* imparted an antioxidant effect via the Nrf2/Keap1/ARE signaling pathway to reduce the ROS and MDA production [35].

In summary, the purified *C. speciosa* antioxidant-enriched extract significantly up-regulated the expressions of Nrf2 and HO-1, while down-regulating the expression of Keap1, thereby protecting the HepG2 cells against oxidative damage. Previous studies on *C. speciosa* reported that its polyphenols were the major components responsible for the antioxidant effect [36]. Other research has linked the antioxidant action of polyphenols with their abilities to monitor oxidative stress, via regulating the Nrf2 signaling pathway, and suppressing the production and reaction of the oxidative mediators and oxidative-related enzymes (e.g., ROS, MDA, GSH-PX, CAT, γ-GCS, HO-1, GCLC and NQO1) [37]. An extract from *C. speciosa* (rich in polyphenols, saponins, oleanolic acid and ursolic acid) was found to exhibit a significant antioxidant activity, as this extract could inhibit the production of MDA, increase the activities of the antioxidant enzymes (e.g., SOD, CAT and GSH-Px), and participate in upregulating the Nrf2/ARE-mediated antioxidant enzymes (HO-1, Trx, GCLM, and GCLC) [38].

## 3. Materials and Methods

### 3.1. Reagents and Materials

Fresh *C. speciosa* fruits were harvested from Linyi, Shangdong province, China in September 2019. Tween 20 and Tween 80 were purchased from BASF Chemical Trade Co., Ltd. (Tianjin, China). The LSA-900C macroporous resin and other resins were provided by Lanxiao Technology New Materials Co., Ltd. (Xi’an, China). Folin–Ciocalteu reagent was provided by Beijing Soleibao Technology Co., Ltd. (Beijing, China); gallic acid, DPPH and ABTS were purchased from Hualan Chemical Technology Co., Ltd. (Shanghai, China); ascorbic acid standard, sodium dodecyl sulfate (SDS), anhydrous ethanol, anhydrous sodium carbonate, sodium hydroxide, concentrated hydrochloric acid, etc. were procured from Tianjin Kaitong Chemical Reagent Co., Ltd. (Tianjin, China). Dulbecco’s Modified Eagle’s Medium (DMEM, C11995500BT), phosphate buffered saline (PBS, C20012500BT), trypsin-EDTA (0.25%) (25200-056) were purchased from ThermoFisher Biochemical Products Co., Ltd. (Beijing, China). Fetal bovine serum (FBS, 900-208) was obtained from Gemini Ltd. (New York, NY, USA). CCK-8 (C0038), the reactive oxygen species (ROS) assay kit (S0033S), malondialdehyde (MDA) assay kit (S0131S), catalase (CAT) assay kit (S0082), total superoxide dismutase (SOD) assay kit (S0088) were all purchased from Beyotime Biotechnology Co., Ltd. (Jiangsu, China).

### 3.2. Extraction of the Antioxidants from C. speciosa Fruits

#### 3.2.1. Single Factor Experiments

Single factor experiments were performed to determine the optimal range of each key variable (factor). *C. speciosa* fruits were dried at 45 °C using a constant temperature oven (Supo Instrument Co., Ltd., Shaoxing, China), pulverized and sieved (60–80 mesh). A quantity (1.0 g) of the fruit powder was accurately weighed, and mixed with 20.0 mL of 50% ethanol solution and 1.0 g/mL different surfactants (Tween 20, Tween 80, SDS, SDBS and CTAB), before the ultrasonic-assisted extraction at 40 °C and an electric power of 200 W for 30 min using an ultrasound processor with a bar (KQ-250DE, Kunshan Ultrasonic Instrument Co., Ltd., Jiangsu, China) [39]. Then, the resulting extract was filtered. The supernatant was collected, and the residue was subjected to the above-described extraction. The extraction process was repeated twice. All of the supernatants were collected, combined, centrifuged at 4800 g and at room temperature for 10 min using a centrifuge (LXJ-IIB, Anting Scientific Instrument Factory, Shanghai, China). Finally, the extract was placed in a 50-mL volumetric flask to allow a constant volume to be achieved. Then, the phenolic yield was calculated by determining the total phenolic content using the Folin–Ciocalteu reagent according to the procedure reported by Cui et al. [40], with a slight modification. The absorbance at 765 nm was measured using a UV spectrophotometer (UV-8000A, Yuanxi, Shanghai, China). The results were expressed as milligram gallic acid equivalent per gram dry weight of *C. speciosa* material (mg GAE/g DW).

According to the above experimental conditions, the single factor experiments were performed by changing one of the factors: the SDS concentration (from 0.2 to 1.2 g/mL), the ethanol concentration (from 30% to 80% *v*/*v*), the material-to-liquid ratio (from 1:10 to 1:35 g/mL), the ultrasonic temperature (from 20 to 70 °C), the ultrasonic time (from 10 to 60 min) and the ultrasonic power (from 100 to 500 W).

#### 3.2.2. Plackett–Burman Design

Plackett–Burman designs are efficient screening designs and allow the identification of the most important factors in the early experimental phase. Based on the results of the single factor experiments, Design Expert 8.0 software (Stat-Ease, Minneapolis, MN, USA) was used for the design and data processing of the Plackett–Burman experiments. Six independent variables at two levels with 12 runs were organized (Table 1). The main effect was evaluated as a difference between the average measurements of each variable, made at a high level (+1) and a low level (−1), which allowed the determination of the optimal extraction conditions. The low level was set as the optimal value obtained from the single factor experiment.

#### 3.2.3. Box–Behnken Design (BBD)

According to the results of the Plackett–Burman test, the extraction parameters, that is, the ethanol concentration, the material-to-liquid ratio and the extraction temperature were optimized using a three-factor-three-level test (Table 2). Following the regression analysis of the data obtained from the experiments, based on the BBD, the second-order polynomial equation was established as follows:(2)$$Y=\beta_{0}+\sum_{i}\beta_{i}X_{i}+\sum_{i}\beta_{\mathrm{ii}}X_{i}^{2}+\sum_{i\neq j}\beta_{\mathrm{ij}}X_{i}X_{j}$$
where: *Y*, the predicted response; *β*_0_, a constant; *β_i_*, *β_ii_* and *β_ij_*, the linear variable coefficient, quadratic variable coefficient, and the coefficient of the three main variables for interaction, respectively; *X_i_* and *X_j_*, the coded independent variables.

### 3.3. Purification of the Antioxidant-Enriched Extract Using the Macroporous Resins

#### 3.3.1. Pretreatments of the Crude Antioxidant-Enriched Extract and the Macroporous Resins

The crude antioxidant-enriched extract, obtained under the optimal extract conditions of the BBD were concentrated by a rotary evaporator, reconstituted in water (concentration1.0 mg/mL) and subjected to further purification using macroporous resins.

Macroporous resins were pretreated, according to a previous method [41]. Briefly, the resins were soaked in a 95% aqueous ethanol solution (*v*/*v*) for 24 h (to allow for the complete swelling), washed with deionized water (till no alcohol smell was detected), soaked in a 5% aqueous HCl solution (*w*/*v*) for 6 h and washed with deionized water till the pH reached neutral. Then, the resins were soaked in a 5% NaOH solution (*w*/*v*) for 6 h, and washed with deionized water till the pH reached neutral.

#### 3.3.2. Static Adsorption and Desorption Tests

A quantity (1.0 g) of each of the different macroporous resins (AB-8, D-101, LX-8, S-8, LSA-900C or LSA-900E) was mixed with 25 mL of the crude antioxidant-enriched extract (1.0 mg/mL) in a 100-mL conical flask. The conical flasks containing the crude extract and each type of resin were continually shaken in a thermostatic oscillator (120 r/min) at 25 °C for 24 h. Following the absorption equilibrium, the resins were filtered and desorbed with 25 mL of a 50% aqueous ethanol solution (*v*/*v*) under continuous shaking (120 r/min, 25 °C for 24 h) [42]. The adsorption rate, desorption rate and recovery rate of each resin, was calculated according to the following equations:
Adsorption rate (%) = [(m_0_ − m_1_)/m_0_] × 100
Desorption rate (%) = [m_2_/(m_0_ − m_1_)] × 100
Recovery rate (%) = (m_2_/m_0_) × 100
(3)

where: m_0_, the initial mass of the antioxidants, mg; m_1_, the equilibrium mass of the antioxidants after adsorption, mg; m_2_, the equilibrium mass of the antioxidants after desorption.

#### 3.3.3. Dynamic Adsorption and Desorption Tests

The LSA-900C resins (10 g), as the selected resins, were wet-loaded onto a glass chromatography column (Ø 0.6 cm × 50 cm), with the adsorption and desorption processes taking place under the experimental conditions. The eluates were collected under the corresponding adsorption and elution conditions, and the mass concentrations of the antioxidants in the eluates were measured [41]. The experimental conditions examined, were as follows: the pH of the crude extract solution (from 2 to 8), the mass concentration of the crude extract solution (from 0.5 to 4.0 mg/mL), the flow rates of the crude extract solution (from 0.5 to 2.5 mL/min), the concentration of the eluent (from 40% to 60%) and the flow rate of the eluent (from 0.5 to 2.5 mL/min).

### 3.4. Chemical- and Cell-Based Antioxidant Capacity Evaluations

#### 3.4.1. Total Reducing Power Assay

The reducing power of the purified antioxidant-enriched extract was determined spectrophotometrically (at 700 nm), according to a previous method [43] with a minor modification. Ascorbic acid was used as a control. The results were expressed as µmol of Fe^2+^ per gram dry weight of *C. speciosa* material (µmol Fe^2+^/g DW).

#### 3.4.2. DPPH Assay

The DPPH scavenging activity was carried out, following a published method [44] with a slight modification. Ascorbic acid chemical was used as a control. The DPPH scavenging ability (%) was determined, based on the absorbance at 517 nm, and evaluated, according to the following equation:Scavenging ability (%) = [1 − (A_1_ − A_2_)/A_3_] × 100
(4)

where: A_1_, 3.0 mL of the test sample solution with 3.0 mL of the DPPH solution; A_2_, 3.0 mL of the test sample solution with 3.0 mL of absolute ethanol; A_3_, 3.0 mL of the DPPH solution with 3.0 mL of 50% aqueous ethanol.

#### 3.4.3. ABTS Assay

The ABTS radical scavenging activity was measured, based on a previous report [44]. Ascorbic acid was used as a control. The ABTS scavenging ability (%) was determined, based on the absorbance at 734 nm and evaluated using the following equation:Scavenging ability (%) = [1 − (A_1_ − A_2_)/A_3_] × 100
(5)

where: A_1_, 0.5 mL of the test sample solution with 3.0 mL of the ABTS working solution; A_2_, 0.5 mL of the test sample solution with 3.0 mL of absolute ethanol; A_3_, 3.0 mL of the ABTS working solution with 0.5 mL of 50% aqueous ethanol.

#### 3.4.4. Cell Culture

The HepG2 cells were obtained from the Chinese Academy of Sciences (Beijing, China). The HepG2 cells were cultured in a fresh high-glucose DMEM medium supplemented with 10% fetal bovine serum and 1% penicillin/streptomycin mixture (100 U/mL penicillin; 100 µg/mL streptomycin) at 37 °C in a humidified atmosphere with 5% CO_2_.

#### 3.4.5. Cell Viability Assay

The HepG2 cells were cultured to 80–90% confluence, then the cells were seeded in a 96-well plate at a density of 5 × 10^4^ cells/mL with 100 µL per well. Following 24 h of culture, the HepG2 cells were treated with the fresh medium containing the purified *C. speciosa* antioxidant-enriched extract at different concentrations (0–100 µg/mL). Then, after further culture for 24 h, an aliquot (10 µL) of the CCK-8 solution was added. The mixture was incubated at 37 °C for 2 h, before the optical density (OD) at 450 nm was measured, using an enzyme labeling instrument (Bio Tek, New England, MA, USA).

#### 3.4.6. Construction of the Oxidative Damage Cell Model

The HepG2 cells were treated, as described in Section 3.4.5. Once the HepG2 cells were cultured for 24 h, the treatment groups were established through pretreating the cells with the purified *C. speciosa* antioxidant-enriched extract (2.5, 5 or 10 µg/mL) for 24 h. Then, these cells were treated with TBHP (0–1000 µg/mL) for 4 h (i.e., subjected to oxidative stress).

Once the HepG2 cells were cultured for 24 h, the blank group and model group were established by culturing the cells with the fresh medium for 24 h. Then, the cells of the model group were further treated with 500 µmol/L TBHP for 4 h.

The cell viability of the blank group, model group and treatment group was measured, based on the absorbance at 450 nm, using the CCK-8 solution, to evaluate the influence of the purified *C. speciosa* antioxidant-enriched extract on the cell viability in the oxidative damage cell model.

#### 3.4.7. Determination of the Biomarkers Associated with the Oxidative Stress and Antioxidant Actions

The HepG2 cells were seeded in 6-well plates at a density of 1 × 10^6^ cells/well and allowed to culture for 24 h. A blank group (the normal HepG2 cells), a model group (HepG2 cells treated with 500 µmol/L TBHP), and a treatment group (HepG2 cells treated with the purified *C. speciosa* extract (2.5, 5 or 10 µg/mL) and TBHP (500 µmol/L)) were set up and subjected to an incubation for 24 h. Then, after the treatments, the total proteins of each group of cells were extracted and quantified using a bicinchoninic acid (BCA) kit (Beyotime, Shanghai, China). Then, the ROS level, MDA content, SOD activity and CAT activity of each group of cells was measured using the ROS kit, MDA kit, SOD kit and CAT kit (Beyotime, Shanghai, China), respectively.

#### 3.4.8. Western Blot Analysis

The HepG2 cells were handled, according to Section 3.4.7. The extracted total protein samples from each group of cells were mixed with the loading buffer, then boiled in boiling water for 10 min to allow for denaturation. The proteins were separated using a 10% sodium dodecyl sulphate-polyacrylamide gel electrophoresis (SDS-PAGE) system and transferred to polyvinylidene fluoride (PVDF) membranes (Millipore, Bedford, London, UK). Then, the membranes were incubated first with 5% nonfat milk at room temperature for 2 h, then incubated at 4 °C overnight with primary antibodies, including Nrf2 (A1244, Abcam, Shanghai, China), Keap1 (A1820, Abcam), HO-1 (A19062, Abcam) and β-actin (GB12001, Servicebio, Wuhan, China). Then, the incubated membranes were washed three times with a Tris-buffered saline containing Tween-20 (TBST, pH 7.4, 0.05%), and incubated with a proper secondary antibody (Goat Anti-Rabbit IgG (ab97051, Abcam) or Goat Anti-Mouse IgG, (ab205719, Abcam)). Finally, the enhanced chemiluminescence (ECL) agent (Beyotime, Biotechnology, Beijing, China) was used to allow the visualization of the target proteins. The densities of the bands were quantified using Image J. β-Actin was used as a reference.

### 3.5. Statistical Analysis

All data were reported as “mean ± standard deviation” of at least triplicate measurements per type of sample. A one-way analysis of variance (ANOVA) and Duncan’s test were conducted to significant differences between the data, using the SPSS 17.0 (IBM, Chicago, IL, USA), and the *p* value < 0.05 was considered statistically significant. The charts were created by Microsoft Excel, and the densities of the bands were quantified using Image J.

## 4. Conclusions

In this study, an antioxidant-enriched extract was obtained for the first time from *C. speciosa* via a surfactant-mediated ultrasonic-assisted process. Based on the single-factor experiments, the Plackett–Burman design and the Box–Behnken design, the optimum extraction conditions were obtained as follows, leading to the maximum phenolic yield of 32.42 mg/g: SDS concentration, 0.40 g/L; ethanol concentration, 58%; material-to-liquid ratio, 1:33 g/mL; ultrasonic treatment, 51 °C, 200 W, 30 min. The crude phenolic extract was further purified using one of five types of macroporous resins. Through the static adsorption and desorption tests, as well as the dynamic adsorption and desorption tests, the optimum purification conditions were obtained: LSA-900C macroporous resin; sample concentration, 2.0 mg/mL; pH, 3; loading flow rate, 1.0 mL/min; loading volume, 100 mL; ethanol concentration, 50%; elution flow rate, 1.5 mL/min; eluting volume, 120 mL. Following the purification with the LSA-900C macroporous resin under the optimum conditions, the content of phenolics reached 57% (8.78 times as high as that of the crude phenolic extract), indicating that phenolic compounds were the major antioxidants in the purified extract. The results of the total reducing power, DPPH and ABTS scavenging assays showed that both the crude extract and the purified extract possessed significant in vitro antioxidant activities, with the purified being more potent. In addition, the cell-based antioxidant capacity evaluation, the antioxidant potential of the purified extract were further confirmed in a TBHP-stimulated HepG2 cell model. The antioxidant actions of the purified extract included the increases of the activities of the SOD and CAT and decreases of the ROS and MDA productions, through regulating the oxidative stress genes and proteins associated with the Nrf2/Keap1 signaling pathway. These findings suggest that *C. speciosa* is rich in natural antioxidants, and it is feasible to produce an antioxidant-enriched extract from the *C. speciosa* fruit. The extract contains up to 57% phenolic compounds and exhibits significant antioxidant effects.

## Figures and Tables

**Figure 1 molecules-27-07970-f001:**
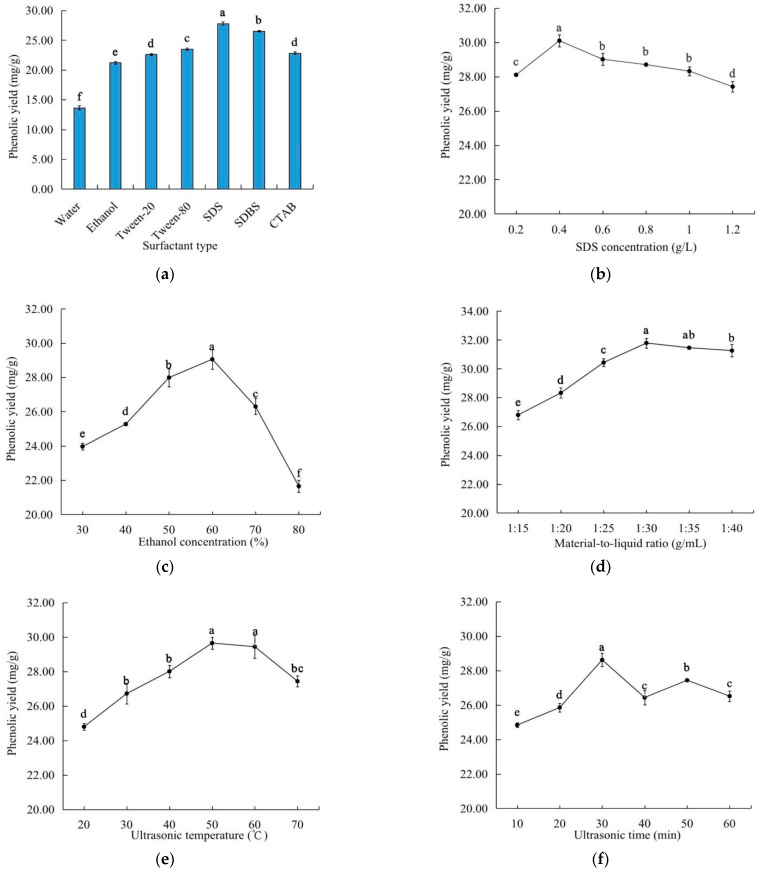
Effects of the different variables on the extraction yield of the total phenolics: (**a**) surfactant type; (**b**) SDS concentration; (**c**) ethanol concentration; (**d**) material-to-liquid ratio; (**e**) ultrasonic temperature; (**f**) ultrasonic time; (**g**) ultrasonic power. Different letters indicate significant differences (*p* < 0.05).

**Figure 2 molecules-27-07970-f002:**
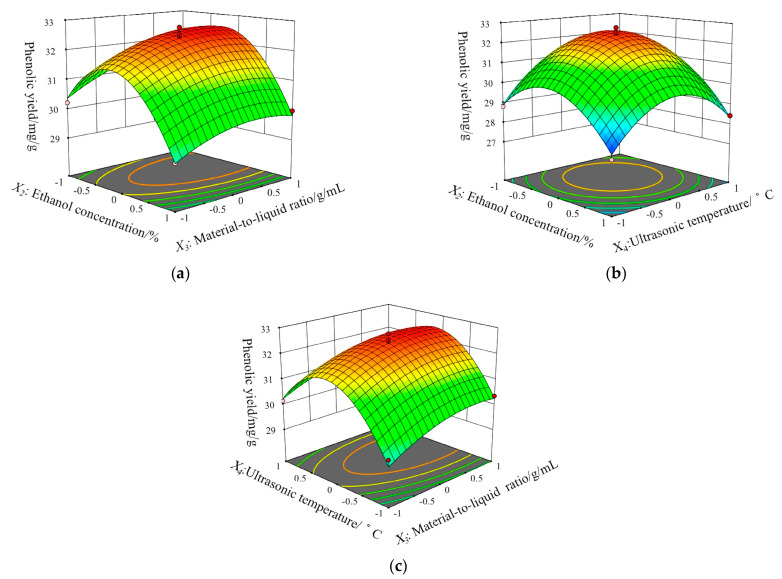
Response surface plots showing the effect of two of the three independent variables on the extraction yield (mg/g) of the total phenolics, the two changed variables were: (**a**) ethanol concentration and the material-to-liquid ratio; (**b**) the ethanol concentration and ultrasonic temperature; (**c**) the material-to-liquid ratio and ultrasonic temperature.

**Figure 3 molecules-27-07970-f003:**
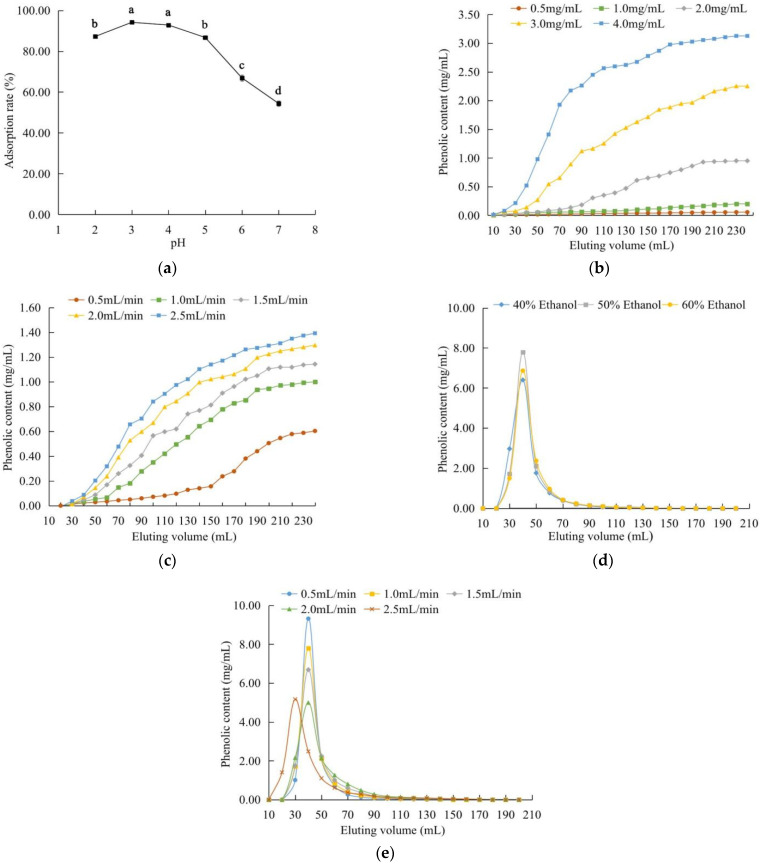
Effects of the different variables on the adsorption and desorption properties of LSA-900C: (**a**) the pH of the crude extract solution; (**b**) the mass concentration of the crude extract solution; (**c**) the flow rates of the crude extract solution; (**d**) the concentration of ethanol in the eluent; (**e**) the flow rate for the elution. Different letters indicate significant differences (*p* < 0.05).

**Figure 4 molecules-27-07970-f004:**
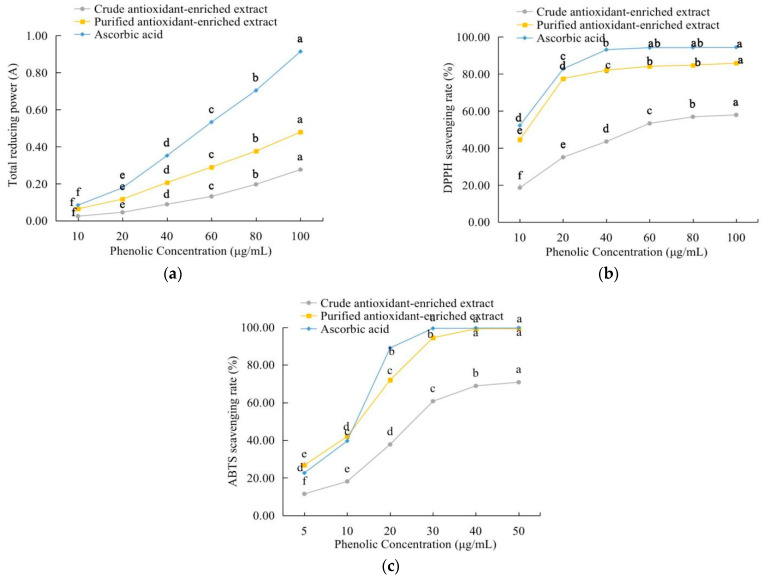
Antioxidant capacities of the *C. speciosa* antioxidant-enriched extract: (**a**) the total reducing power; (**b**) DPPH radical-scavenging activity; (**c**) ABTS radical-scavenging activity. Different letters indicate significant differences (*p* < 0.05).

**Figure 5 molecules-27-07970-f005:**
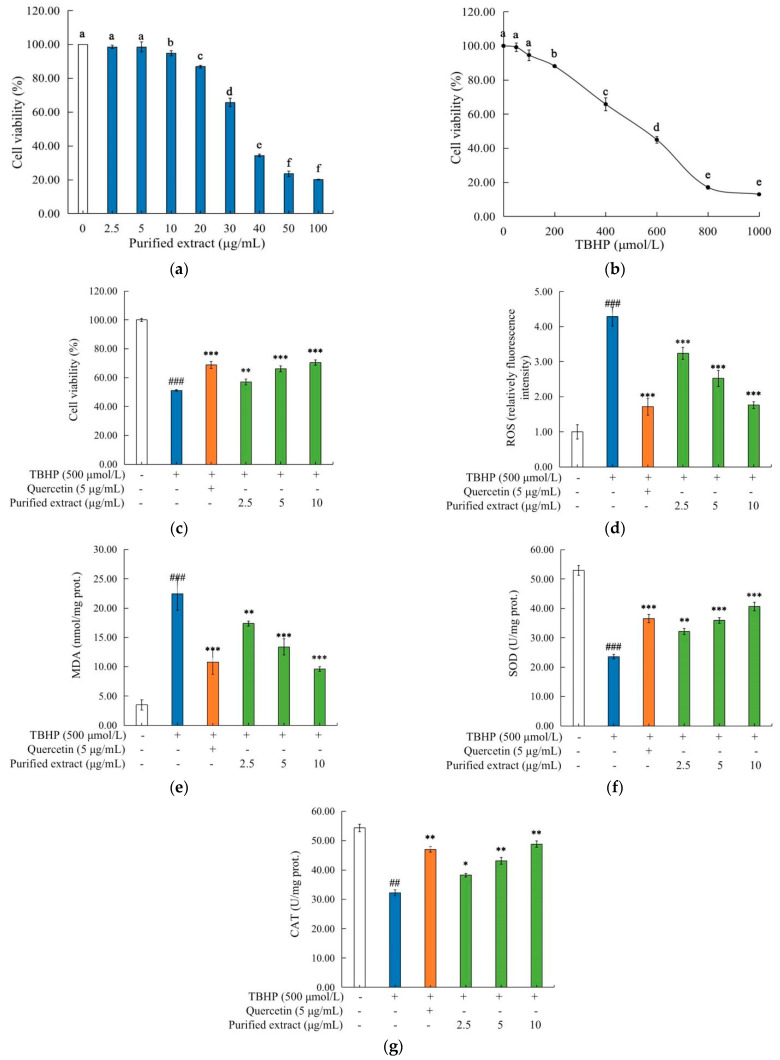
Effects of the purified *C. speciosa* antioxidant-enriched extract on the TBHP-induced injury in HepG2 cells: (**a**) influence of the concentration of the purified *C. speciosa* antioxidant-enriched extract on cell viability; (**b**) influence of the TBHP concentration on cell viability; (**c**) concentration-dependent protective effects of the purified *C. speciosa* antioxidant-enriched extract; (**d**) reactive oxygen species (ROS) level; (**e**) malondialdehyde (MDA) level; (**f**) superoxide dismutase (SOD) level; (**g**) catalase (CAT) level; (**h**) expressions of Nrf2, HO-1 and keap1. The data are shown as means (column heights) ± standard deviations (error bars) (*n* = 3 independent observations). Different letters indicate significant differences (*p* < 0.05); ## *p* < 0.01, ### *p* < 0.001 compared with the control group; * *p* < 0.05, ** *p* < 0.01, *** *p* < 0.001 compared with the TBHP-treated group.

**Table 1 molecules-27-07970-t001:** Plackett–Burman experimental design and results.

Run	Coded Variable Levels						Phenolic Yield (mg/g)
	X_1_	X_2_	X_3_	X_4_	X_5_	X_6_	
1	−1 (0.2)	1 (70)	1 (1:35)	−1 (40)	1 (40)	1 (300)	28.82 ± 0.21
2	−1 (0.2)	1 (70)	−1 (1:25)	1 (60)	1 (40)	−1 (100)	27.82 ± 0.26
3	−1 (0.2)	1 (70)	1 (1:35)	1 (60)	−1 (20)	−1 (100)	28.79 ± 0.30
4	1 (0.6)	1 (70)	1 (1:35)	−1 (40)	−1 (20)	−1 (100)	28.09 ± 0.27
5	1 (0.6)	1 (70)	−1 (1:25)	1 (60)	1 (40)	1 (300)	27.14 ± 0.37
6	1 (0.6)	−1 (50)	1 (1:35)	1 (60)	−1 (20)	1 (300)	32.48 ± 0.38
7	1 (0.6)	−1 (50)	1 (1:35)	1 (60)	1 (40)	−1 (100)	31.62 ± 1.29
8	−1 (0.2)	−1 (50)	−1 (1:25)	1 (60)	−1 (20)	1 (300)	30.02 ± 0.43
9	−1 (0.2)	−1 (50)	1 (1:35)	−1 (40)	1 (40)	1 (300)	31.25 ± 0.80
10	−1 (0.2)	−1 (50)	−1 (1:25)	−1 (40)	−1 (20)	−1 (100)	29.04 ± 0.34
11	1 (0.6)	1 (70)	−1 (1:25)	−1 (40)	−1 (20)	1 (300)	25.66 ± 0.38
12	1 (0.6)	−1 (50)	−1 (1:25)	−1 (40)	1 (40)	−1 (100)	28.04 ± 0.16

Note: X_1_ = SDS concentration (g/L), X_2_ = Ethanol concentration (%), X_3_ = Material-to-liquid ratio (g/mL), X_4_ = Ultrasonic temperature ( ℃), X_5_ = Ultrasonic time (min), X_6_ = Ultrasonic power (W).

**Table 2 molecules-27-07970-t002:** The effect value of each factor and its significance analysis.

Source	Sum of Squares	df	Mean Square	F Value	*p*-Value Probe > F
Model	0.48	6	0.080	14.44	0.0051
X_1_-X_1_	2.437 × 10^−3^	1	2.437 × 10^−3^	0.44	0.5360
X_2_-X_2_	0.24	1	0.24	44.27	0.0012 **
X_3_-X_3_	0.17	1	0.17	30.98	0.0026 **
X_4_-X_4_	0.053	1	0.053	9.58	0.0270 *
X_5_-X_5_	1.268 × 10^−4^	1	1.268 × 10^−4^	0.023	0.8855
X_6_-X_6_	7.351 × 10^−3^	1	7.351 × 10^−3^	1.33	0.3008
Residual	0.028	5	5.523 × 10^−3^		
Cor Total	0.51	11			

Note: X_1_ = SDS concentration (g/L), X_2_ = Ethanol concentration (%), X_3_ = Material-to-liquid ratio (g/mL), X_4_ = Ultrasonic temperature (℃), X_5_ = Ultrasonic time (min), X_6_ = Ultrasonic power (W); Level of significance: * *p* < 0.05, ** *p* < 0.01.

**Table 3 molecules-27-07970-t003:** Box–Behnken design and results.

Number	Coded Variable Levels			Phenolic Yield (mg/g)
	X_2_	X_3_	X_4_	
1	0 (60)	0 (1:30)	0 (50)	32.23 ± 0.58
2	−1 (50)	−1 (1:25)	0 (50)	30.21 ± 0.88
3	−1 (50)	0 (1:30)	1 (60)	30.41 ± 0.68
4	−1 (50)	1 (1:35)	0 (50)	31.54 ± 0.80
5	0 (60)	−1 (1:25)	1 (60)	30.05 ± 0.34
6	1 (70)	0 (1:30)	1 (60)	28.42 ± 0.36
7	1 (70)	0 (1:30)	−1 (40)	27.58 ± 0.37
8	0 (60)	1 (1:35)	−1 (40)	30.35 ± 0.80
9	0 (60)	1 (1:35)	1 (60)	30.78 ± 0.59
10	0 (60)	−1 (1:25)	−1 (40)	29.45 ± 0.89
11	0 (60)	0 (1:30)	0 (50)	32.46 ± 0.45
12	1 (70)	−1 (1:25)	0 (50)	29.15 ± 0.32
13	−1 (50)	0 (1:30)	−1 (40)	28.81 ± 0.99
14	0 (60)	0 (1:30)	0 (50)	32.55 ± 0.32
15	1 (70)	1 (1:35)	0 (50)	29.98 ± 0.17
16	0 (60)	0 (1:30)	0 (50)	32.05 ± 0.58
17	0 (60)	0 (1:30)	0 (50)	32.47 ± 0.45

Note: X_2_ = Ethanol concentration (%), X_3_ = Material-to-liquid ratio (g/mL), X_4_ = Ultrasonic temperature ( ℃).

**Table 4 molecules-27-07970-t004:** Evaluation of the regression equation model coefficients and their significance test.

Source	Sum of Squares	df	Mean Square	F Value	*p*-Value Probe > F
Model	0.38	9	0.043	47.48	<0.0001 ***
X_2_	0.043	1	0.043	47.92	0.0002 ***
X_3_	0.017	1	0.017	19.13	0.0033 **
X_4_	0.016	1	0.016	17.91	0.0039 **
X_2_X_3_	6.250 × 10^−4^	1	6.250 × 10^−4^	0.70	0.4296
X_2_X_4_	1.444 × 10^−3^	1	1.444 × 10^−3^	1.62	0.8433
X_3_X_4_	1.822 × 10^−4^	1	1.822 × 10^−4^	0.20	0.6645
X_2_^2^	0.13	1	0.13	150.74	<0.0001 ***
X_3_^2^	6.983 × 10^−3^	1	6.983 × 10^−3^	7.85	0.0265 *
X_4_^2^	0.14	1	0.14	157.14	<0.0001 ***
Residual	6.228 × 10^−3^	7	8.897 × 10^−4^		
Lack of Fit	3.091 × 10^−3^	3	1.030 × 10^−3^	1.31	0.3861
Fure Error	3.137 × 10^−3^	4	7.842 × 10^−4^		
Cor Total	0.39	16			
R^2^ = 0.9840	R^2^_adj_ = 0.9634	C.V. = 0.98			

Note: X_2_ = Ethanol concentration (%), X_3_ = Material-to-liquid ratio (g/mL), X_4_ = Ultrasonic temperature (℃); Level of significance: * *p* < 0.05, ** *p* < 0.01, *** *p* < 0.001.

**Table 5 molecules-27-07970-t005:** Effects of the different types of macroporous resins on the adsorption and desorption of the phenolics.

Resin Type	Adsorption Capacity (mg/g)	Adsorption Rate (%)	Desorption Rate (%)	Recovery Rate (%)
AB-8	21.50 ± 0.47 ^c^	78.83 ± 1.05 ^c^	89.51 ± 3.24 ^a^	70.57 ± 3.03 ^b^
D101	22.94 ± 0.63 ^b^	82.73 ± 1.27 ^c^	84.07 ± 2.88 ^b^	69.56 ± 2.93 ^b^
LX-8	23.85 ± 0.96 ^ab^	87.64 ± 2.05 ^b^	29.37 ± 2.67 ^d^	25.71 ± 2.01 ^d^
S-8	24.89 ± 0.67 ^a^	88.70 ± 1.50 ^b^	1.60 ± 1.63 ^e^	1.50 ± 1.45 ^e^
LSA-900C	25.19 ± 0.28 ^a^	90.72 ± 0.81 ^a^	84.84 ± 4.22 ^b^	76.95 ± 3.13 ^a^
LSA-900E	25.15 ± 0.33 ^a^	90.20 ± 0.47 ^ab^	71.48 ± 0.06 ^c^	64.44 ± 0.28 ^b^

Note: Different letters indicate significant differences (*p* < 0.05).

## Data Availability

Not applicable.

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
