# Peer review of "Surfactant-Mediated Ultrasonic-Assisted Extraction and Purification of Antioxidants from Chaenomeles speciosa (Sweet) Nakai for Chemical- and Cell-Based Antioxidant Capacity Evaluation"

_molecules, 2022, doi:10.3390/molecules27227970_

Round 1

Reviewer 1 Report

L11 Please establish the need of new sources of antioxidants

L12 Could you write the country or region where  Chaenomeles is popular

L13 Please add the part of the plant you used to sonicate

L93 How do you explain that?

L109 What do you mean with “substances”, may be phenols? How do you prove that? So you see some precipitate?, please explain. If phenols precipitate, they cann’t be detected in solution.

L112 When you say  that “the material-to-liquid ratio peaked at the ratio of 1:30 g/mL” I don’t agree because there is no difference between 1:30 and 1:35 g/mL, please correct.

When you say  “These results indicate 112 that a higher ratio of solvents could accelerate the infiltration of the solvent into the C. 113 speciosa cells” I don’t agree because at 1:40 g/mL ratio it’s a higher ratio, however, the phenolic yield is lower.

L120 In the statement “These results might be due to the fact that a higher temperature, power or time led to i) the oxidation and degradation  of some phenolic compounds”, did you do some experiments to prove that? You could use gallic acid in known concentration, in a similar solution  and in the conditions of temperature, power and time to quantify the concentration before and after exposition.

Table 1 Did you do the experiments by triplicate?, If so, please add the standard deviation of the obtained media value of phenolic yield.

Table 3 idem

Figure 2, you can use the contour or the response surface plots

L131 Please discuss why these three factors had significant influences on the extraction of total phenolics. There are a lot of literature about the effect of this factor on the phenols extraction yield.

Figure 4, please explain the X axis, do you mean phenol concentration or extract concentration, please explain.

L415 Please erase the last C. speciosa

L446 Change capital letters in Constant Temperature Oven

L463 Brand of software

Reviewer 2 Report

The manuscript deals with the extraction and purification of antioxidant substances from Chaenomeles Speciosa. The authors performed a comprehensive study that included the evaluation of different parameters such as surfactant type and concentration, ultrasound application time and power, ethanol concentration, type of resin in the purification step, and finally antioxidant capacity and action using different methodologies.  Overall the study has solid scientific soundness and is well-written and presented. I only suggest some improvement in the editing and description of the methodology.

Specific comments are listed below:

Details of the ultrasound processor should be specified, including if ultrasounds were applied with a probe or a bath.

In sentences 554-555 a word is probably missing.

Line 556-557, how were the model and blank group established?

Line 572 the assay kit should be specified

There is no information about tween 20 and tween 80 in the reagents and materials section.

The authors do not describe how the experiments of different surfactants were carried out. This is the first result presented but there is no information in the methodology, for example, which concentration was used?

The single-factor experiments were performed to evaluate each parameter (SDS concentration, ethanol concentration, material-to-liquid ratio, ultrasonic temperature, ultrasonic time and ultrasonic power) I assume that the rest of the parameters were fixed but there is not any information about which conditions were used. For example, which ethanol concentration was used when the surfactant concentration was evaluated?

The tools and software used for the mathematical modelling and the graphical representation should be specified.

Line 188 This is an average? The deviation should be added

Line 617 57 % is a dry weight value?
